# Differential Effects of Selenium Compounds on Mitochondrial Function in PRRSV-Infected Porcine Alveolar Macrophages

**DOI:** 10.3390/v17101303

**Published:** 2025-09-26

**Authors:** Abigail Williams, Christina Bourne, John Byrne, Chaitawat Sirisereewan, Brittany M. Pecoraro, Elisa Crisci

**Affiliations:** Department of Population Health and Pathobiology, College of Veterinary Medicine, North Carolina State University, Raleigh, NC 27607, USA; aewill25@ncsu.edu (A.W.); ckbourne@ncsu.edu (C.B.); csirise@ncsu.edu (C.S.); brittanypecoraro@students.rossu.edu (B.M.P.)

**Keywords:** selenium, swine, nutrition, PRRSV, alveolar macrophages, mitochondria

## Abstract

Selenium (Se) is a trace mineral with antioxidant and anti-inflammatory properties. Se deficiency increases oxidative stress and immunosuppression. In swine, dietary Se supplementation enhances immunity and growth, and previous studies suggest it protects immune cells during viral infection. Porcine reproductive and respiratory syndrome virus (PRRSV) causes severe respiratory and reproductive failure in swine, resulting in annual losses of 1.2 billion USD. Vaccine efficacy is hampered by the virus’s high mutation rate, requiring alternative approaches. This study examines the effects of organic (DL-Selenomethionine, L-Selenomethionine, yeast-selenium) and inorganic (sodium selenite) Se on PRRSV infection in vitro. Porcine alveolar macrophages, the primary target of PRRSV in the lung, were isolated from healthy animals and infected with PRRSV-2 with or without Se. Mitochondrial function, gene expression, oxidative stress, and viral load were assessed post-infection. DL-selenomethionine showed increased glycolytic and mitochondrial ATP production relative to other compounds, suggesting improved mitochondrial function. No antiviral activity against PRRSV was observed. Transcriptome analysis revealed infection-driven modulation, with upregulation of IL6, IL8, IL1B1, MX1, and TXNRD1, but Se had no significant effect. While Se did not exhibit antiviral activity in vitro, its enhancement of mitochondrial function offers additional insight supporting its potential immunomodulatory benefits observed in previous in vivo studies.

## 1. Introduction

Porcine reproductive and respiratory syndrome virus (PRRSV) is one of the most detrimental diseases for the global swine industry, causing severe respiratory distress and reproductive failure. In the US alone, PRRSV results in an economic loss of 1.2 billion USD annually [1,2]. PRRSV is an enveloped positive-stranded RNA virus belonging to the family *Arteriviridae* and order *Nidovirales*. PRRSV circulates globally as two species, *Betaarterivirus europensis* (European origin) and *Betaarterivirus americense* (North American origin) [3,4,5]. Swine are the only natural host of PRRSV, which has high tissue tropism to cells of the monocyte lineage, particularly porcine alveolar macrophages (PAMs) and pulmonary interstitial macrophages [6,7,8]. Alveolar macrophages are pivotal components of the lung’s immune defense against respiratory viral pathogens, preserving pulmonary integrity and cellular homeostasis [9,10,11]. Therefore, dysregulation of macrophage function compromises antiviral responses and contributes to heightened immunosuppression and inflammation.

PRRSV infection hinders both the innate and adaptive immune responses by selectively targeting immune cells, which is associated with incomplete viral clearance. The virus disrupts immune signaling and antigen presentation, and this immune modulation correlates with prolonged viral shedding [7,12]. Additionally, PRRSV infection impedes mitochondrial biogenesis and induces oxidative stress to facilitate viral proliferation and immune evasion [13]. Disruptions in cellular homeostasis and mitochondrial function are key components in PRRSV pathogenesis, complicating disease control efforts [13,14,15]. Commercially available modified live vaccines are widely used for disease control, yet have limited cross-protection due to the high mutation rate of the virus and potential for reversion to virulence and recombination [13,16]. The result is a need for alternative strategies for mitigating PRRSV burden in the field.

One approach involves nutritional immunomodulation through supplementation with trace minerals and vitamins. Selenium (Se) is an essential trace mineral known for its anti-inflammatory, antioxidant, and potential antiviral properties. Through its biological function as selenoproteins, specifically selenocysteine, Se is involved in the regulation of oxidative stress and modulation of the immune response [17]. Antioxidant selenoenzymes, such as glutathione peroxidases (GPXs) and thioredoxin reductases (TrxRs), protect cells against reactive oxygen species (ROS) and nitric oxide (NO) overproduction [18,19]. Biological systems are constantly exposed to intracellular or environmentally produced free radicals, which are minimized through the activity of such antioxidant proteins [20,21]. A previous study has demonstrated that zinc oxide–Se nanoparticles alleviate PRRSV-induced oxidative stress through the antioxidant properties of selenium, which facilitates restoring redox balance [22]. Similarly, in porcine circovirus type 2 infections (PCV2), selenomethionine treatment restored oxidative dysfunction caused by increased ROS levels and reduced viral DNA copies by up to 95% [23], supporting the hypothesis that selenium possesses qualities capable of mitigating virus-induced oxidative damage and reducing viral replication. 

Selenium is routinely included in commercial swine feed between 0.15 and 0.3 mg/kg to improve growth performance and support a variety of physiological functions [17,24]. A deficiency in dietary selenium is associated with increased virulence, progression of viral infections, and decreased antioxidant activity in swine [25,26]. In chickens infected with low pathogenic avian influenza H9N2, yeast-selenium and sodium selenite were incorporated into dietary feed at commercial feed concentrations, and Se-treated animals had reduced cloacal and oropharyngeal viral shedding and increased IFNA, IFNB, and IFNG expression when compared to untreated counterparts [27]. These findings suggest dietary Se may enhance antiviral response across multiple host species and provide a dietary supplementation option to modulate PRRSV infection.

Building on the use of Se in commercial swine production and its well-characterized antioxidant and anti-inflammatory properties, the objective of this study was to evaluate the in vitro immunomodulatory and antiviral effects of Se compounds in PAMs upon PRRSV infection. We tested four different Se sources: DL-Selenomethionine (DL), L-Selenomethionine (L), yeast-based selenium (Yeast), and sodium selenite (NaSe). Specifically, this study investigates the capacity of different Se compounds to restore mitochondrial function, modulate the NO and ROS produced during oxidative stress, and reduce PRRSV load. This work aims to assess the different formulations of selenium and their influence on cellular metabolism and oxidative stress in PAMs, evaluating the molecular mechanism behind shaping the host immune response during PRRSV infection.

## 2. Materials and Methods

### 2.1. Cells and Selenium Compounds

Monkey African Green Kidney (MA-104) (ATCC, CRL-2378.1T) cells were cultured in DMEM supplemented with 10% heat-inactivated fetal bovine serum (FBS), 100 μg/mL streptomycin, 100 IU/mL penicillin, 2 nM L-glutamine, and 10 μg/mL gentamicin in a 37 °C, 5% CO_2_ incubator.

PAMs were isolated from the lungs of influenza A virus and PRRSV-negative pigs. PAM cells were obtained through bronco alveolar lavage (BAL) as described previously [28,29]. PAM cells were frozen in 1 mL aliquots of 10% dimethyl sulfoxide/FBS and stored in liquid nitrogen. PAMs were cultured in RPMI media, supplemented with 10% heat-inactivated FBS, 100 μg/mL streptomycin, 100 IU/mL penicillin, 2 nM L-glutamine, and 10 μg/mL gentamicin in a 37 °C, 5% CO_2_ incubator. 

Four different selenium compounds were evaluated: organic DL-selenomethionine (Catalog No. 240775000, Thermo Fisher Scientific, Waltham, MA, USA), L-selenomethionine (Catalog No. 259962500, Thermo Fisher Scientific, Waltham, MA, USA), inorganic sodium selenite (Catalog No. 194741, MP Biomedicals, Solon, OH, USA), and a proprietary organic yeast-derived selenium. Selenium powder compounds were weighed and diluted in cell culture water (Catalog No. 25-055-CV, Corning, Corning, NY, USA) and filter sterilized using 0.22 µm syringe filters. Compounds were prepared to 0.3 parts per million (ppm) concentration from a stock 1 mg/mL solution. Assays were performed at 0.3 ppm or 0.03 ppm concentrations. 

### 2.2. Viruses

This study utilized one North Carolina PRRSV-2 isolate, NC134 (GenBank accession ID ON844087), lineage 1C.3 [30,31]. Virus stocks were propagated and titrated using MA-104 cell lines and stored in aliquots at −80 °C as described previously [28,32].

### 2.3. PRRSV-2 In Vitro Infection Layout

PAMs were washed and seeded in a 96-well flat-bottomed plate at 150,000 cells per well and incubated at 37 °C for a minimum of 2 h. After incubation, the supernatant was removed from each well. Cells were infected with the virus at a multiplicity of infection (MOI) of 0.5, containing Se compounds at 0.3 ppm or 0.03 ppm concentrations. After treatment, cells were incubated in a 37 °C incubator for an additional 2 h. After 2 h, the supernatant was removed from each well and treated with Se compounds at 0.3 ppm or 0.03 ppm concentrations, or RPMI alone. After an additional 22 h of incubation, the cell supernatant was collected for titration (TCID_50_ method), and cell lysate was collected using Trizol^TM^ lysis buffer (Catalog No. 15596026, Thermo Fisher Scientific, Waltham, MA, USA) or lysis buffer (Catalog No. 100015634, Ambion, Carlsbad, CA, USA) for RNA extraction and downstream RT-qPCR analysis.

### 2.4. Cytotoxicity Assay

The cytotoxicity of Se compounds in PAM was determined by assessing cell membrane integrity using the CellTox^TM^ Green Cytotoxicity Assay kit following the endpoint Method (Catalog No. G8741, Promega, Madison, WI, USA). This assay quantifies the fluorescent signal proportional to the asymmetric cyanine dye binding to dead cells’ DNA. Briefly, PAMs were washed and seeded at 150,000 cells per well in a 96-well flat-bottomed plate with black sides (Greiner Bio-one, Monroe, NC, USA). The plate was incubated at 37 °C for 2 h. After adhesion, PAMs were treated with Se at 0.3 and 0.03 ppm concentrations. Untreated PAMs were used as a control. PAMs were incubated with Se compounds for a total of 24 h at 37 °C and a positive control (lysis solution) was added to specific wells in the last 2 h. Green fluorescent dye was added to all wells and incubated at room temperature for an additional 15 min, protected from light. The assay was read using a spectrophotometer (BioTek, Winooski, VT, USA) at an excitation wavelength of 485–500 nm and emission of 520–530 nm. The results are reported as Relative Fluorescent Units (RFU).

### 2.5. Tissue Culture Infectious Dose (TCID_50_) Analysis

Supernatant collected from PRRSV in vitro infection (Section 2.3) was used for virus titration, as described previously [33]. Titers were calculated via the Spearman–Kärber method [34].

### 2.6. RT-qPCR

Cell lysate was collected with Trizol^TM^ lysis buffer (Catalog No. 15596026, Thermo Fisher Scientific, Waltham, MA, USA) or lysis buffer (Catalog No. 100015634, Ambion, Carlsbad, CA, USA) according to manufacturer instructions. RNA was extracted and purified using Zymo RNA MiniPrep (Catalog No. R2052, Zymo Research, Irvine, CA, USA) or Qiagen RNAEasy Mini Kit (Catalog No. 52904, Qiagen, Hilden, Germany) according to the manufacturer’s instructions. Nanodrop technology was used to evaluate RNA quantity and purity. cDNA synthesis was performed using the Applied Biosystems High-Capacity cDNA Reverse Transcription Kit (Catalog No. 4368814, Thermo Fisher Scientific, Waltham, MA, USA). Extracted RNA was normalized to 1.0 ng/µL prior to cDNA synthesis.

Relative mRNA expression was evaluated by qPCR using the iQ SYBR Green Supermix (Catalog No. 1708882, Bio-Rad, Hercules, CA, USA). Samples were evaluated in the Applied Biosystems Step OnePlus Real Time PCR Instrument (Thermo Fisher Scientific, Waltham, MA, USA). qPCR cycling conditions were 95 °C for 3 min, and were linked to 40 cycles of 95 °C for 10 s and 60 °C for 30 s. Relative gene expression was calculated following the ΔCt method. The primers used to detect PRRSV-2 strain NC-134 were nsp9 F (5′CCTGCAATTGTCCGCTGGTTT-3′) and nsp9 R (5-GACGACAGGCCACCTCTCTTAG-3′) [29]. Ribosomal protein S24 (RPS24) was used as a reference gene for normalization, as previously described [30]. All primers were purchased from Integrated DNA Technologies (Durham, NC, USA). 

### 2.7. Evaluation of Mitochondrial Fitness

Oxygen Consumption Rate (OCR) and Extracellular Acidification Rate (ECAR) of PRRSV-2 and Se compound-treated PAMs were measured at 37 °C using Seahorse Extracellular Flux Analyzer, with ATP Real Time Test and Cell Mito Stress Test (Agilent, Santa Clara, CA, USA). Before the testing day, cells were washed and counted, then seeded at 300,000 cells/150 µL per well in the testing cartridge (Catalog No. 103022-100, Agilent, Santa Clara, CA, USA) and incubated for 2 h at 37 °C. After 2 h adhesion, PAMs were treated with or without PRRSV-2 MOI 0.5 and Se at 0.03 ppm for an additional 22 h. Untreated cells were used as the control. On testing day, detection cartridges (Catalog No. 103022-100, Agilent, Santa Clara, CA, USA), Seahorse media, and different drugs were prepared following the manufacturer’s protocol (Catalog No. 103591-100, Agilent, Santa Clara, CA, USA). For the ATP Real Time Test, oligomycin (Oligo) and rotenone/AA (RA) were used at final concentrations in the well of 1.5 µM and 0.5 µM, respectively. For the Cell Mito Stress Test, Oligo, carbonyl cyanide-4 (trifluoromethoxy) phenylhydrazone (FCCP), and RA were used at final concentrations in the well of 1.5 µM for Oligo, 2 µM for FCCP, and 0.5 µM for RA. Measurements were performed on the Agilent Seahorse XFp analyzer (Agilent, Santa Clara, CA, USA), available at North Carolina State University, College of Veterinary Medicine. Glycolytic ATP production and mitochondrial ATP production were analyzed for the ATP assay; basal respiration, maximal respiration, non-mitochondrial oxygen consumption, and spare respiratory capacity were analyzed for the Cell Mito Stress. All analyses were performed using Wave Desktop 2.6 software (Agilent).

### 2.8. Measurement of Reactive Oxygen Species (ROS) and Nitric Oxide (NO)

To measure ROS, PAMs were washed and seeded at 150,000 cells per well with RPMI without phenol red in a 96-well flat-bottomed black plate (Catalog No. 655090, Greiner Bio-one, Monroe, NC, USA). The plate was incubated for 2 h in a 37 °C incubator. After adhesion, cells were treated with PRRSV-2 MOI 1.0 and Se compounds at 0.03 ppm concentration. Dihydrorhodamine was added to cells at a concentration of 10 µM, following the manufacturer’s instructions (Catalog No. D0134, Chemodex, St. Gallen, Switzerland). The plate was incubated for 15 min at 37 °C protected from light and read at different time points (0 h, 12 h, 24 h, 48 h, 72 h, and 96 h) using the BioTek Synergy 2 Plate Reader (BioTek, Winooski, VT, USA) at an excitation wavelength of 485–500 nm and emission of 520–530 nm. The results are reported as ROS Production in RFU. 

The measurement of nitrite concentration as an index of NO production was analyzed using the Griess Reagent System following the manufacturer’s instructions (Catalog No. G2930, Promega, Madison, WI, USA). Briefly, PAMs were washed and seeded at 150,000 cells per well with RPMI without phenol red on a 96-well flat-bottomed plate. The plate was incubated for 2 h in a 37 °C incubator. After adhesion, cells were treated with PRRSV-2 MOI 1.0 and Se compounds at 0.03 ppm concentration. The plate was incubated for 24 h in a 37 °C incubator. Supernatants were collected at 24 h post-infection, and a nitrate standard curve was prepared following the manufacturer’s instructions. The supernatants and nitrate standard curve were added to a new 96-well flat-bottomed plate, and treated with sulfanilamide solution for 10 min, and then naphthyl ethylenediamine dihydrochloride solution for an additional 10 min to form a stable azo compound. The plate was read using BioTek Synergy 2 plate reader (BioTek, Winooski, VT, USA) at an excitation wavelength of 485–500 nm and emission of 520–530 nm. The results are reported as NO concentration (µM).

### 2.9. Gene Expression Analysis via NanoString Technology

Total RNA was extracted from PRRSV-2 and Se compound-treated PAMs using Zymo RNA MiniPrep (Catalog No. R2052, Zymo Research, Irvine, CA, USA) or Qiagen Viral RNA Mini Kit (Catalog No. 52904, Qiagen, Hilden, Germany) following the manufacturer’s instructions. Nanodrop technology was used to evaluate the extracted RNA quantity and purity. Extracted RNA was normalized between 9 and 20 ng/µL prior to hybridization. NanoString nCounter (NanoString Technologies, Seattle, WA, USA) was utilized following the manufacturer’s instructions. Panels are customized to quantify 36 target genes, 10 housekeeping genes (Appendix A), and additional positive and negative internal controls. Briefly, normalized RNA was hybridized in a thermocycler for 18 h at 65 °C, with a panel-specific reporter and capture probe set. Hybridized samples were then loaded and processed on the nCounter Prep Station, where the samples were immobilized onto a cartridge. The cartridge was then scanned using the nCounter Digital Analyzer (NanoString Technologies, Seattle, WA, USA) at 555 fields of view resolution. 

Raw digital count data were exported from the nCounter Digital Analyzer and uploaded to nSolver Software (Version 4.0) for quality control (QC). Samples that failed QC were excluded from downstream analysis. Gene expression counts were normalized to housekeeping genes and background thresholding. Normalized data were exported in raw counts, and a fold change was calculated compared to the control and shown as arbitrary units (AU). 

### 2.10. Statistical Analysis

Statistical analysis was performed using GraphPad Prism 10.4.0 (GraphPad Software, San Diego, CA, USA). Agilent Seahorse Cell Mito Stress Test and ATP Real Time Test results were analyzed using Wave Software (Agilent, Santa Clara, CA, USA). NanoString nCounter results were analyzed using nSolver Software Version 4.0 (NanoString Technologies, Seattle, WA, USA). The statistical methods for each assay are listed in the respective figure captions.

## 3. Results

### 3.1. Selenium Compounds Exhibit Concentration-Dependent Cytotoxicity in PAMs

Organic and inorganic Se cytotoxicity was measured in PAMs at 0.3 ppm and 0.03 ppm concentrations after 24 h treatment to determine which concentration of Se induces cytotoxicity. All Se compounds at 0.3 ppm concentration had statistically significantly increased cytotoxic activity when compared to the mock (Figure 1). There were no statistically significant differences observed in the 0.03 ppm concentration treatment when compared to the mock **(Figure 1**). 

Based on the cytotoxicity results in Figure 1, a concentration of 0.03 ppm was selected for all subsequent experiments in PAMs, as the level of cytotoxicity was similar to that of the untreated PAM culture (mock).

### 3.2. DL-Selenomethionine Treatment Modulates ATP Production in PRRSV-2 Infected Macrophages and Restores Mitochondrial Function

To evaluate the metabolic function and ATP production of PRRSV-2 NC134 infected and Se compound-treated PAMs, we measured the Oxygen Consumption Rate (OCR) for oxidative phosphorylation and the ATP production rate using Seahorse technology (Figure 2A). OCR values for mock and NC134+DL exposed PAMs demonstrated similar metabolic responses, whereas NC134, NC134+L, and NC134+NaSe showed a substantial decrease in OCR levels (Figure 2A). We observed a significant decrease in mitochondrial function across the NC134, NC134+L, and NC134+NaSe treatment groups, also supported by the decreased glycolytic and mitochondrial ATP production levels when compared to the mock (Figure 2A). In contrast, DL-selenomethionine treatment restored both glycolytic and mitochondrial ATP Production to mock treatment levels (Figure 2A). DL-selenomethionine-treated PAMs maintained similar mitochondrial activity as untreated cells (Figure 2A), indicating a restoration in energy production pathways.

Additionally, we assessed basal respiration, maximal respiration, non-mitochondrial respiration, and spare respiratory capacity of NC134 and Se compound-treated PAMs to assess mitochondrial function (Figure 2B). We observed statistically significant decreases in all the NC134-Se treatment group OCR values compared to untreated PAMs for all respiration parameters. Data suggest that Se treatment was not capable of reversing the shift towards glycolytic pathways versus oxidative phosphorylation observed during PRRSV infection in vitro (Figure 2B). 

### 3.3. PRRSV-Associated Oxidative Stress Persists Despite Se Treatment in PAMs

NO and ROS production evaluated the influence of Se compound treatments on PRRSV-associated oxidative stress. At 24 h, the PRRSV infection increased the NO production in PAMs when compared to the mock (Figure 3A). No statistically significant differences in NO levels were observed in infected PAMs treated with Se compounds (Figure 3A). Interestingly, infected PAMs exposed to yeast treatment showed a significant increase in NO levels compared to control PAMs.

We did not find any statistically significant differences in ROS production in PRRSV-infected and Se compound-treated PAMs when compared to mock across 96 h (Figure 3B). ROS values increased substantially across all treatments after 48 h yet yielded no differences between treatment groups at the end of the assay.

### 3.4. Transcriptomic Data upon Se Treatment of Infected PAMs Reveals Limited Modulation of Immune Cellular Responses

Transcriptome expression analysis of antioxidant enzymes, cytokines, and intracellular signaling genes revealed significant increases in these genes during infection and Se treatment (Figure 4). As expected, viral infection led to a marked upregulation of inflammatory cytokines, such as IL6, IL8, IL1B1, and IFNG, together with TGFB, indicating a strong immunomodulation to infection (Figure 4C–G). MX1, an antiviral cellular response factor, was also upregulated upon viral infection (Figure 4H). When selenium treatments were combined with infection, they kept the same immunomodulatory profile as PRRSV infection alone (Figure 4).

GPX4 and TXNRD1, selenoprotein antioxidant genes responsible for maintaining cellular homeostasis and protection against oxidative stress and hydroperoxides, demonstrated limited modulation after infection (Figure 4A). We observed statistical significance with all selenium treatments when paired with infection (Figure 4A). Expression of TXNRD1 was significantly upregulated in infected PAMs compared to mock, but no significant differences were observed between DL, L, and NaSe treatments (Figure 4B).

Additional genes involved in inflammatory responses, immune cell activation, cell surface receptors, and innate immune effectors were analyzed during infection with or without DL, L, and NaSe selenium treatments (Appendix A). Infected PAMs exposed to yeast selenium (Appendix A) followed a similar immunomodulatory pattern as seen in Figure 4.

### 3.5. Se Treatment Fails to Confer Antiviral Protection in PRRSV-2 Infected PAMs

Preliminary virus titration by TCID_50_ assay in the MA-104 cell line using DL, NaSe, and yeast Se compounds showed a statistically significant reduction in viral titers after DL and yeast treatments at 0.3 ppm (Appendix A). In PAMs, the TCID_50_ assay yielded no statistically significant decrease in viral titer between DL, L, and NaSe selenium treatments (Figure 5A). Similarly, RT-qPCR analysis in PAMs revealed no differences in viral load after selenium treatment (Figure 5B).

## 4. Discussion

In this study, we investigated the in vitro effects of different selenium compounds on mitochondrial function, oxidative homeostasis, and antiviral responses in porcine alveolar macrophages upon PRRSV-2 infection. First, we evaluated the cytotoxic effects of different concentrations of organic and inorganic Se to best determine which concentration was most suitable for immune modulation without impairing cell viability. An in vitro model using porcine lymphocytes treated with NaSe showed decreased apoptosis and cell death with treatment at 0.15 ppm [14,35]. In our study, each Se compound exhibited substantial cell death in primary macrophages at 0.3 ppm compared to the 0.03 ppm treatment, highlighting that 0.3 ppm represents the upper limit for in vitro cell viability.

Mitochondrial dysfunction is increasingly recognized as a hallmark of viral pathogenesis, driven by virus-induced mitophagy and excessive mitochondrial fission to facilitate viral replication [36]. Such mitochondrial impairment has been well-documented in Hepatitis B Virus, Influenza, and SARS-CoV-2 systems [20,21,37]. We examined ATP Production and Cell Mito Stress pathways and noted comparable trends of mitochondrial impairment upon PRRSV infection, consistent with previous studies using PAMs and MARC-145 cell line to assess mitochondrial fitness [15,38]. DL-selenomethionine restored ATP production through both glycolytic and mitochondrial pathways and remained in a restoration state under the influence of ATP synthase inhibitors. Our result aligns with previous findings of selenium treatment preserving mitochondrial respiratory chain complex activities under oxidative stress conditions in viral, bacterial, and hypoxic systems [23,39,40]. However, we did not observe this effect with L-selenomethionine, NaSe, or yeast-selenium compounds. These differences highlight a limitation in the bioactivity of different selenium compounds and suggest that the chemical form of selenium plays a critical role in modulating the mitochondrial function of primary macrophages.

Given the established role of mitochondrial dysfunction in promoting oxidative stress, we evaluated the cellular-derived NO and ROS as a marker of redox imbalance during PRRSV infection. Viral infection in the lung triggers an inflammatory response and recruits immune cells, both of which contribute to oxidative stress [41,42]. PRRSV induces oxidative stress in infected macrophages, with oxidative mediators such as NO and ROS playing key roles in the host’s inflammatory and antiviral response [43,44,45]. Selenium, through its biological function as selenocysteine, mediates redox imbalance by maintaining membrane integrity [45,46]. In HIV-1 and Influenza virus systems, Se treatment inhibited virus activation via oxidative stress production after in vitro treatment between 0.025 and 0.2 ppm, respectively [47,48]. In contrast to previous in vitro findings [23,49], Se treatment did not demonstrate any significant reduction in NO and ROS levels relative to NC134 infection. Although NO production did not significantly differ between mock and selenium treatment conditions, DL-selenomethionine showed a trend (*p* = 0.061) towards reduced NO levels between virus and DL-selenomethionine. These findings suggest that Se’s redox modulating effects may be insufficient to counteract virus-induced oxidative stress in this model.

PRRSV infection is well documented to suppress both innate and adaptive immune responses by dysregulating key signaling pathways, specifically those governed by interferons and cytokines [50,51]. Transcriptome analysis of PRRSV infected myeloid cells revealed changes in cell movement and cell to cell signaling pathways, confirming that these biological pathways play pivotal roles in viral infection [50,52]. Similarly to previous gene expression studies [50,53], PRRSV upregulated inflammatory cytokines (IL8, IL6, IL1B, TNFA) and interferons (IFNA, IFNB) in our study. Conversely, although previous research has demonstrated that Se can modulate pro-inflammatory factors [46], we did not observe such modulation after Se treatment under PRRSV-infected conditions. We also examined the expression of antioxidant selenoproteins (i.e., GPX4 and TXNRD1), which reduce hydroperoxides and thioredoxins to regulate free radical metabolism [54,55,56]. In contrast with selenium’s role in upregulating antioxidant enzymes to promote redox homeostasis [57,58]. Se treatment decreased the transcriptional activity of GPX4 after infection. However, TXNRD1 followed a similar pattern as observed with the inflammatory cytokines. In vivo yeast–selenium treatment has been previously described to modulate the suppression of IL2, IFNG, IL1B, and TNFA expression after Aflatoxin and PCV2 exposure, respectively [54,59]. In the present study, in vitro yeast-selenium treatment revealed similar findings in the immune gene transcriptional activity as observed in DL-selenomethionine, L-selenomethionine, and NaSe compound treatment. The variability in cellular responses observed in the transcriptome analysis is likely driven by the inherent differences in primary cells and subsequent variability in selenium uptake by the cells. Primary alveolar macrophages exhibit donor-dependent differences in baseline activation state, gene expression, and responsiveness to external stimuli, which may underline the observed variability in the analysis [60].

Selenium treatment has numerous protective effects across various viral systems [23,27,61], primarily through interference with viral replication dynamics and attenuation of downstream immune activation [55,56]. However, in our study using PAMs, the primary target cell for PRRSV, we found that Se treatment did not reduce viral titer or load when tested at 0.03 ppm. Aligning with prior in vitro studies involving Se treatment in immortalized cell lines [23,49], we observed a reduction in viral titer in MA-104 cells at 0.3 ppm Se treatment; suggesting dose-dependent or cell type limitations to its antiviral capacity in vitro. This discrepancy between the efficacy of Se treatment to reduce viral replication in cell lines and primary cell systems may be attributed to the differences in Se uptake by the cell, cell-specific antiviral response, and overall cell health after storage conditions.

## 5. Conclusions

In conclusion, the present study demonstrated that although Se treatment did not reduce PRRSV-2 NC134 viral replication in porcine alveolar macrophages, the organic DL-selenomethionine exerted beneficial effects on mitochondrial function, and the potential for reducing oxidative stress. Through preservation of metabolic pathways during PRRSV infection, DL-selenomethionine may ameliorate cellular stress independent of its effect on viral replication in PAM. These findings highlight DL-selenomethionine as a promising candidate for supplementary molecular studies, as well as further in vivo investigation in swine health and production.

## Figures and Tables

**Figure 1 viruses-17-01303-f001:**
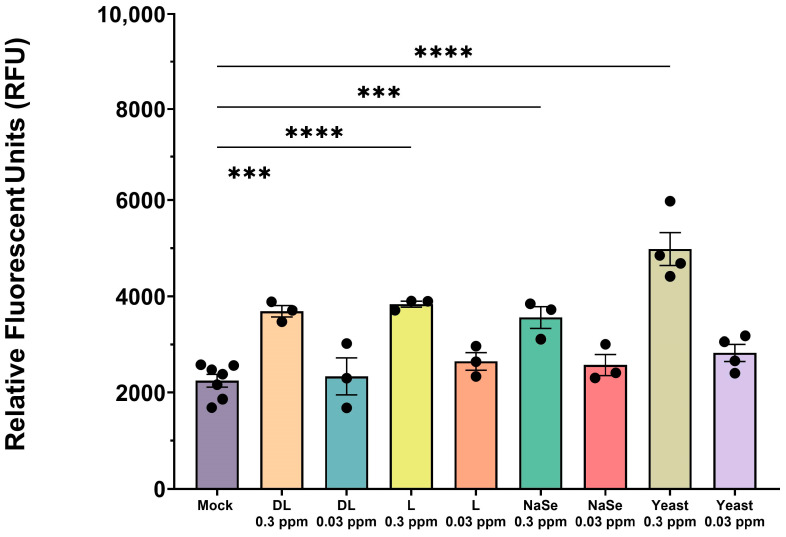
Cytotoxicity assay of PAMs exposed to Se compounds at 0.3 ppm and 0.03 ppm concentrations. Cells were treated with organic or inorganic Se, or left untreated (mock) for 24 h. All experiments were performed using three technical replicates in three–four biological replicates for each treatment; mock (*n* = 7). Each symbol represents one biological replicate. Statistical significance was determined by comparing the means of all independent groups using One Way ANOVA with Tukey’s multiple comparison test. Results displayed showing the mean and standard error of the mean (SEM). *** *p* ≤ 0.001, **** *p* ≤ 0.0001.

**Figure 2 viruses-17-01303-f002:**
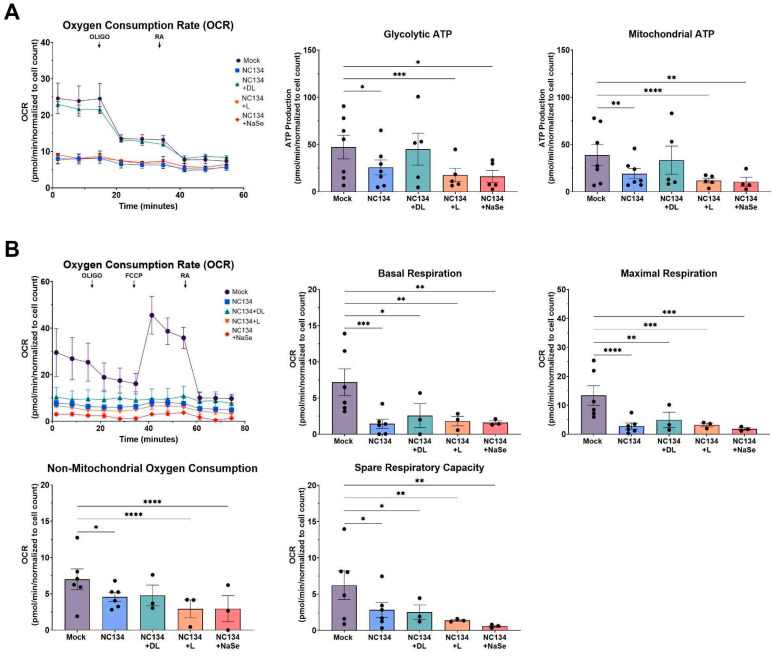
Mitochondrial fitness analysis of PRRSV-2 NC134 and Se compound-treated PAMs. Cells were infected with NC134 MOI 0.5, treated with organic or inorganic Se at 0.03 ppm, or left untreated (mock) for 24 h. PAMs were exposed sequentially to Oligo and RA for ATP Real Time Assay; and Oligo, FCCP, and RA for Cell Mito Stress Test. (**A**) ATP Real Time Assay parameters, (**B**) Cell Mito Stress Respiration Parameters. The assays were performed using two technical replicates in five–seven biological replicates for each treatment. Each symbol represents one biological replicate. Statistical significance was determined by comparing the means of all independent groups using a Two Way ANOVA with Dunnett’s multiple comparison test. Results displayed showing the mean and standard error of the mean (SEM). * *p* ≤ 0.05, ** *p* ≤ 0.01, *** *p* ≤ 0.001, **** *p* ≤ 0.0001.

**Figure 3 viruses-17-01303-f003:**
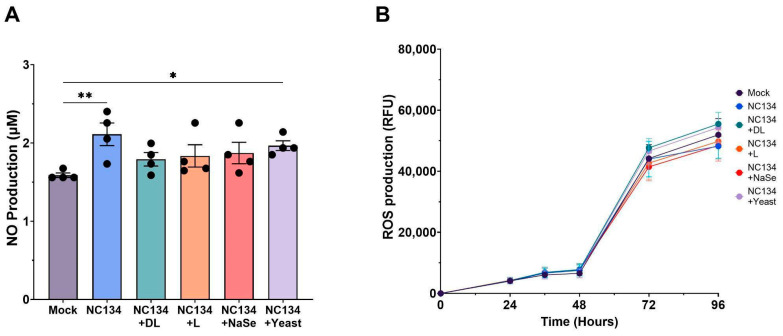
NO and ROS Production of PAMs upon PRRSV-2 NC134 and Se compound treatments. Cells were infected with NC134 MOI 0.5, treated with organic or inorganic Se at 0.03 ppm, or left untreated (mock). (**A**) NO Production and (**B**) ROS Production, visualized across 96 h. All experiments were performed using between three and four technical replicates in four biological replicates for each treatment. Each symbol represents one biological replicate. Statistical significance was determined by comparing the means of all independent groups using One Way ANOVA. Results displayed showing the mean and standard error of the mean (SEM). * *p* ≤ 0.05, ** *p* ≤ 0.01.

**Figure 4 viruses-17-01303-f004:**
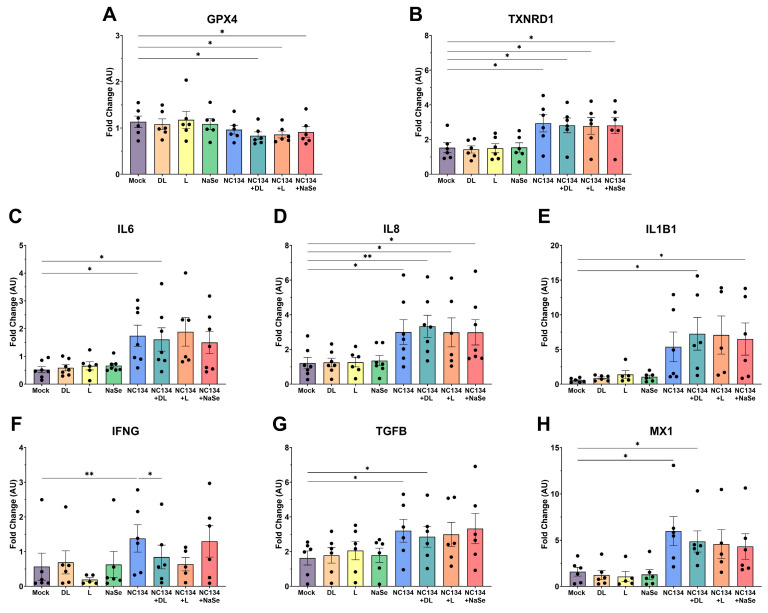
Gene expression of immune responses in PRRSV-2 NC134 and Se compound-treated PAMs, using NanoString technology. PAMs were infected with NC134 MOI 0.5, treated with organic or inorganic Se at 0.03 ppm, or left untreated (mock). Obtained values were normalized to the mean of mock values and shown as Fold Change (AU). Fold changes in (**A**) GPX4, (**B**) TXNRD1, (**C**) IL6, (**D**) IL8, (**E**) IL1B1, (**F**) IFNG, (**G**) TGFB, (**H**) MX1. All experiments were performed using between six and seven biological replicates. Each symbol represents one biological replicate. Statistical significance was determined by comparing the means of all independent groups using One Way ANOVA. Results displayed showing mean and standard error of the mean (SEM). * *p* ≤ 0.05, ** *p* ≤ 0.01.

**Figure 5 viruses-17-01303-f005:**
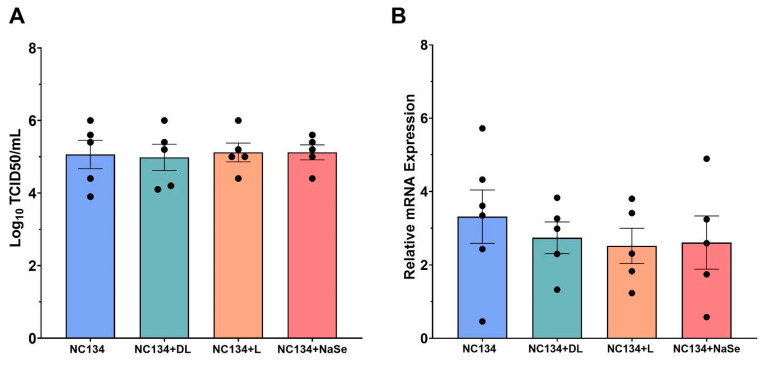
Measurement of the antiviral capacity of PRRSV-2 NC134 and Se compound-treated PAMs. Briefly, cells were infected with NC134 at an MOI of 0.5 and treated with organic or inorganic Se at 0.03 ppm. Supernatant collected for TCID_50_ analysis, and cell lysate collected for RT-qPCR. (**A**) Log_10_ TCID_50_ titers upon NC134 infection and Se treatment at 0.03 ppm. Experiments were performed using five technical replicates in five biological replicates. (**B**) Relative mRNA expression upon NC134 infection and Se treatment at 0.03 ppm. Experiments were performed using three technical replicates between five and six biological replicates. Each symbol represents one replicate. Statistical significance was determined by comparing the means of all independent groups using One Way ANOVA. Results displayed showing mean and standard error of the mean (SEM).

## Data Availability

The original contributions presented in this study are included in the article/Appendix A. Further inquiries can be directed to the corresponding author.

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
