# Peer review of "Differential Effects of Selenium Compounds on Mitochondrial Function in PRRSV-Infected Porcine Alveolar Macrophages"

_viruses, 2025, doi:10.3390/v17101303_

Round 1
Reviewer 1 Report
Comments and Suggestions for Authors
This paper determined the effects of several different organic or inorganic selenium complexes on Mitochondrial function, gene expression, oxidative stress, and viral load caused by PRRSV infection. The research is highly attractive to readers.
1. The abstract should contain corresponding data, at least important data for support.
2. There is ambiguity in the objects of statistical comparison in Figures 1 and 2. Are the comparisons between two, three, or several objects? Please check and clarify. It is recommended to adopt one-to-one comparisons.
3. The title of 3.1 is inappropriate, and it is suggested to remove the expression of 0.03ppm concentration.
4. The annotation of "To assess the in vitro cytotoxic effects of Se compounds in PAM, the CellToxTM Green Cytotoxicity Assay kit (Catalog No. G8741, Promega, Madison, WI, USA) was used following manufacturer’s instructions" in 2.4 is inappropriate. It should state what method was used to determine what index. Please revise it.
Author Response
See attachment.
Reviewer 1
This paper determined the effects of several different organic or inorganic selenium complexes on Mitochondrial function, gene expression, oxidative stress, and viral load caused by PRRSV infection. The research is highly attractive to readers.
- The abstract should contain corresponding data, at least important data for support.
The abstract has been changed accordingly.
- There is ambiguity in the objects of statistical comparison in Figures 1 and 2. Are the comparisons between two, three, or several objects? Please check and clarify. It is recommended to adopt one-to-one comparisons.
In Figure 1, multiple comparisons were made between all groups to determine if there was a cytotoxic difference between untreated and treated cells. We performed a One Way ANOVA with Tukey’s multiple comparison test as a statistical analysis. We have changed the legend accordingly in lines 265-267.
In Figure 2, multiple comparisons were made between all groups to determine if there was a difference in the respiration parameters between untreated and treated cells. We performed a Two Way ANOVA with Dunnett’s multiple comparison test as a statistical analysis. We have changed the legend accordingly in lines 297-299.
Additionally, we have updated the remaining figure legends in the main text and supplementary information to specify the objects compared for each statistical analysis to clear any ambiguity for the reader.
- The title of 3.1 is inappropriate, and it is suggested to remove the expression of 0.03ppm concentration.
We thank the reviewer for this important point and have updated the title to: “Selenium compounds exhibit concentration-dependent cytotoxicity in PAM”.
- The annotation of "To assess the in vitro cytotoxic effects of Se compounds in PAM, the CellToxTM Green Cytotoxicity Assay kit (Catalog No. G8741, Promega, Madison, WI, USA) was used following manufacturer’s instructions" in 2.4 is inappropriate. It should state what method was used to determine what index. Please revise it.
We thank the reviewer for the comment. The CellToxTM Green Cytotoxicity Assay Kit measures changes in cell membrane integrity that is a result of cellular death. The assay utilizes a proprietary asymmetric cyanine dye that stains dead cells’ DNA and is excluded from live cells. We measured the fluorescent signal produced by the dye binding to compromised cells to observe the cytotoxicity of Se compounds in PAM. We used the endpoint method after 24 hr exposure to Se compounds, allowing for a direct comparison between treated and mock cells, consistent with the assay protocol. We changed the text accordingly in lines 136-141.

Reviewer 2 Report
Comments and Suggestions for Authors
the authors of this manuscript have investigated the possible role of selenium (Se) in preventing or ameliorating the effects of porcine reproductive and respiratory syndrome virus in a cell culture model. The authors report that at a level of Se that does not cause cell death there was some improvement in mitochondria function but no effect on the reproduction of the the virus. Interestingly there appears to be the assumption that the effects of the virus are primarily due to mitochondrial dysfunction and not directly due to the action of the virus death or dysfunction. Thus, improving mitochondrial function with Se might improve the outcome from the infection. I am not convinced that this is the case but it is a possibility. The data supports there conclusions but there is one piece of data that they do not really address. In the results section looking at the effects of Se on cytokines and immunomodulators they do not address the apparent biphasic response shown by Se on Il1B1 and IFN gamma. In the data of both these analyses there appear to be two distinct responses. Some of the cells respond positively and some do not appear to respond at all. The authors should discuss the significance of this in discussion.
Author Response
Reviewer 2
the authors of this manuscript have investigated the possible role of selenium (Se) in preventing or ameliorating the effects of porcine reproductive and respiratory syndrome virus in a cell culture model. The authors report that at a level of Se that does not cause cell death there was some improvement in mitochondria function but no effect on the reproduction of the the virus. Interestingly there appears to be the assumption that the effects of the virus are primarily due to mitochondrial dysfunction and not directly due to the action of the virus death or dysfunction. Thus, improving mitochondrial function with Se might improve the outcome from the infection. I am not convinced that this is the case but it is a possibility. The data supports there conclusions but there is one piece of data that they do not really address. In the results section looking at the effects of Se on cytokines and immunomodulators they do not address the apparent biphasic response shown by Se on Il1B1 and IFN gamma. In the data of both these analyses there appear to be two distinct responses. Some of the cells respond positively and some do not appear to respond at all. The authors should discuss the significance of this in discussion.
We thank the reviewer for this useful comment. Please see lines 432-436 in the discussion for the revised section.

Reviewer 3 Report
Comments and Suggestions for Authors
The paper describes investigation of Se influnece onto PRRSV-2 NC134 viral replication in porcine alveolar macrophages. The authors found no statistic influence on PRRSV replication in vitro. The Se seems to enhance mitochondrial function under PRRSV infection what may suggest DL-selenomethionine as a promising candidate for supplementary molecular studies. Although the paper seems scientifcally clear the final conclusions and the purpose of this study is not fully explained.
Author Response
Reviewer 3
The paper describes investigation of Se influnece onto PRRSV-2 NC134 viral replication in porcine alveolar macrophages. The authors found no statistic influence on PRRSV `replication in vitro. The Se seems to enhance mitochondrial function under PRRSV infection what may suggest DL-selenomethionine as a promising candidate for supplementary molecular studies. Although the paper seems scientifcally clear the final conclusions and the purpose of this study is not fully explained.
We thank the reviewer for this important comment. The text has been updated accordingly in lines 84-87, and 452-457.
